# Nectin Family Ligands Trigger Immune Effector Functions in Health and Autoimmunity

**DOI:** 10.3390/biology12030452

**Published:** 2023-03-15

**Authors:** Doryssa Hermans, Lisa van Beers, Bieke Broux

**Affiliations:** 1University MS Center, Campus Diepenbeek, 3590 Diepenbeek, Belgium; doryssa.hermans@uhasselt.be (D.H.); lisa.vanbeers@student.maastrichtuniversity.nl (L.v.B.); 2Department of Immunology and Infection, Biomedical Research Institute, University of Hasselt, 3590 Diepenbeek, Belgium

**Keywords:** Nectin, Necl, DNAM-1, CRTAM, Tactile, TIGIT, autoimmunity, inflammation, immunological synapse, cytotoxicity

## Abstract

**Simple Summary:**

Immune-cell activation is triggered upon antigen or target recognition. This is modulated by co-stimulatory or co-inhibitory signals provided by the cellular environment. The balance between activating and inhibitory receptors controls immune homeostasis, while a disbalance contributes to chronic inflammation and autoimmunity. Binding partners of the Nectin protein family of adhesion molecules are widely expressed in the immune system and are mainly described in cancer immunology. This review summarizes the expression pattern of Nectin binding partners as immune-activating and -inhibitory receptors and how they modulate immune responses in health and autoimmunity.

**Abstract:**

The superfamily of immunoglobulin cell-adhesion molecules (IgCAMs) is a well-known family of cell-adhesion molecules used for immune-cell extravasation and cell–cell interaction. Amongst others, this family includes DNAX accessory molecule 1 (DNAM-1/CD226), class-I-restricted T-cell-associated molecule (CRTAM/CD355), T-cell-activated increased late expression (Tactile/CD96), T-cell immunoreceptor with Ig and ITIM domains (TIGIT), Nectins and Nectin-like molecules (Necls). Besides using these molecules to migrate towards inflammatory sites, their interactions within the immune system can support the immunological synapse with antigen-presenting cells or target cells for cytotoxicity, and trigger diverse effector functions. Although their role is generally described in oncoimmunity, this review emphasizes recent advances in the (dys)function of Nectin-family ligands in health, chronic inflammatory conditions and autoimmune diseases. In addition, this review provides a detailed overview on the expression pattern of Nectins and Necls and their ligands on different immune-cell types by focusing on human cell systems.

## 1. Introduction

Immune responses of the adaptive immune system are initially triggered when T lymphocytes encounter antigen (Ag)-presenting cells (APCs), primarily dendritic cells (DCs), in secondary lymphoid organs. The T cell–APC contact, i.e., interaction of the T-cell receptor (TCR) and major histocompatibility complex (MHC) molecule, evolves into a signaling cluster of (co-)stimulatory and inhibitory receptors, and adhesion molecules (e.g., lymphocyte-function-associated antigen 1 (LFA-1)), designated as the immunological synapse (IS) [1,2]. Here, protein tyrosine kinases, such as Lck and Fyn, are recruited to the TCR complex to phosphorylate immunoreceptor tyrosine-based activation motifs (ITAMs) [3]. Ultimately, the IS enables long-lasting cell contacts that promote immune-cell activation, proliferation and differentiation [1,2]. Similarly, an IS is formed between cytotoxic cells and their target cells, where cytolytic granules are released inside the synaptic cleft [1,4]. The balance between activating and inhibitory receptors controls immune homeostasis, while a disbalance contributes to chronic inflammation and autoimmunity.

Members of the superfamily of immunoglobulin cell-adhesion molecules (IgCAMs), including DNAX accessory molecule 1 (DNAM-1/CD226), class-I-restricted T-cell-associated molecule (CRTAM/CD355), T-cell-activated increased late expression (Tactile/CD96) and T-cell immunoreceptor with Ig and ITIM domains (TIGIT), are widely expressed within the innate and adaptive immune systems (Table 1). They function as cell-adhesion molecules as well as stimulatory or inhibitory receptors at the IS, by interacting with their ligands of the Nectin and Necl protein family, consisting of four Nectins (Nectin-1 to -4) and five Necls (Necl-1 to -5). Although Nectins and Necls are often cited using alternative nomenclature, depending on the research field (summarized in Table 1), their structural and functional properties are similar [5].

As part of the IgCAM superfamily, they structurally contain extracellular Ig-like domains which determine their ligand binding and affinity: DNAM-1 and CRTAM contain two extracellular domains, while Tactile and TIGIT contain three and one Ig-like domain(s), respectively [6,7]. However, their intracellular domains contain different binding motifs for signaling molecules, which explains their diverse effector functions. DNAM-1 functions as a co-stimulatory receptor and carries an immunoglobulin tail tyrosine (ITT)-like phosphorylation motif, which can boost ITAM-induced signaling [3,8]. Tactile and TIGIT both contain an ITT-like motif as well as an immunoreceptor tyrosine-based inhibition motif (ITIM), enabling inhibition of ITAM-induced signaling [3,9,10]. In contrast, CRTAM contains a PDZ binding motif (PSD-95/Discs-large/ZO-1) [11].

In this review, we provide an overview of the expression profile of the IgCAM protein family within the innate and adaptive immune systems, focusing on human cells. Finally, the immune-cell effector functions triggered by DNAM-1, CRTAM, Tactile and TIGIT will be described in physiological conditions and chronic inflammation.

## 2. Immune-Cell Expression

Cells of the innate and adaptive immune systems, including T and B lymphocytes, NK cells and myeloid cells, broadly express DNAM-1, CRTAM, Tactile and TIGIT (Table 1). In the T-lymphocyte population, T-regulatory cells (Tregs), CD4^+^ T-helper cells (Th) and CD8^+^ cytotoxic T lymphocytes (CTLs) equally express DNAM-1, presenting similar expression levels in the naive, central memory (CM) and effector memory (EM) subset, while Tactile is predominantly expressed by EM cells [12,13,14]. In addition, one third of resting T lymphocytes co-express Tactile and DNAM-1, which are upregulated after TCR-mediated activation [12]. A small subset of B cells expresses DNAM-1 depending on the stage of B-cell maturation [12,15,16,17]. DNAM-1 expression is concentrated in memory B cells, plasmablasts and plasma cells and is upregulated after CpG stimulation, suggesting that it acts as an activation and maturation marker [16]. Autoimmunity affects DNAM-1 expression in various ways whereas an altered Tactile expression is only recently reported in ankylosing spondylitis patients (Table 2) [18]. In multiple sclerosis (MS), DNAM-1 is downregulated on natural-killer (NK) cells and associated with impaired NK-mediated regulation of T-cell activity [19], while it is upregulated on Tregs of progressive MS patients, which can influence their suppressive capacity (Table 2, functional properties are further discussed below) [20]. In systemic sclerosis (SSc), EM CD8^+^ T cells show an upregulation of DNAM-1 which is associated with disease severity, increased cytokine production and cytotoxicity, while its expression is downregulated on NK cells in a subgroup of patients (Table 2) [17,21]. Moreover, single-nucleotide polymorphisms (SNPs) in the DNAM-1 gene have been associated with increased vulnerability to autoimmune diseases, such as MS, rheumatoid arthritis (RA), type 1 diabetes (T1D), systemic lupus erythematosus (SLE), primary immune thrombocytopenia, juvenile idiopathic arthritis and autoimmune thyroid disease [22,23,24,25]. TIGIT is highly expressed by Tregs, followed by memory CD4^+^ and CD8^+^ T lymphocytes [12,26,27]. Recently, TIGIT expression was also identified on memory B lymphocytes exerting a regulatory function [28]. In inflammatory disorders, TIGIT expression is differentially regulated, e.g., TIGIT is upregulated upon TCR triggering or on EM CD4^+^ T cells and Tregs in mild atopic dermatitis, while downregulated on CD4^+^ T lymphocytes in psoriasis patients and severe atopic dermatitis, both negatively correlated with disease severity (Table 2) [26,27,29]. CRTAM is a transient activation marker expressed by activated NK cells, NK T cells, CD8^+^ T lymphocytes and a small subset of CD4^+^ T lymphocytes [30,31,32]. Consequently, its expression is tightly controlled by activating signals such as NK-cell-activating receptor and TCR triggering [30,31,32]. During homeostasis, leukocyte CRTAM expression is restricted (<0.5%) suggesting that an activated immune system or pathological conditions, such as asthma [33] is needed to upregulate its expression [30]. Interestingly, CD4^+^CRTAM^+^ T cells express cytotoxic T-lymphocyte-related genes, indicating that these cells can acquire cytotoxic properties comparable to CD8^+^ T lymphocytes [34,35].

Less is known about the expression of Nectin and Necl proteins themselves within the healthy immune system. While Nectin-3 is the only member expressed by T lymphocytes and NK cells [36,37,38,39], monocytes express a diversity of Nectins and Necls [38,40,41,42,43]. Interestingly, Necl-5 expression is induced on human CD4^+^ T cells after TCR activation, both on naive and memory T lymphocytes, which is reduced in MS patients [19,44]. Since the Nectin and Necl protein family is involved in oncoimmunity by either promoting cancer immunosurveillance or tumor immune escape [45,46,47], their expression can be induced in malignant conditions [48,49,50,51]. For example, ectopic Necl-2 gene expression is induced in patients with adult T-cell leukemia and/or virus -infected T lymphocytes while leukocytes of healthy controls lack Necl-2 expression, even after TCR activation [48,52,53]. However, there is still controversy on Necl-2 protein expression and function on T lymphocytes. While T cells derived from healthy peripheral blood mononuclear cells (PBMCs) show no or low Necl-2 expression [52,53], Jurkat T cells show high expression [54]. Furthermore, Necl-2 is upregulated after TCR-triggering and physically interacts with the TCR in Jurkat T cells, promoting its intracellular signaling [54].

**Table 1 biology-12-00452-t001:** Expression profile of Nectin and Necl proteins and their ligands within the healthy immune system. All data are based on protein expression in human peripheral blood mononuclear cells (PBMCs) or NK-92 and Jurkat T-cell lines. Alternative gene nomenclature was extracted from the National Center for Biotechnology Information (NCBI) and UniProt. CD, cluster of differentiation; HVE, human-herpesvirus entry receptor; PRR/PVRL, poliovirus-receptor-related gene; HIgR, herpesvirus Ig-like receptor; CADM, cell adhesion molecule; SYNCAM, synaptic-cell adhesion molecule; IGSF, immunoglobulin-superfamily gene; TSLL/TSLC, tumor-suppressor gene; TAG, tumor-associated glycoprotein.

	T Lymphocytes	B Lymphocytes	NK Cells	Monocytes Macrophages	Ref.
**DNAM-1** (CD226)	+	+	+	+	[12,13,15,16,17,40,41]
**CRTAM** (CD355)	+	-	+	-	[30,31,33]
**Tactile** (CD96)	+	+/-	+	+	[12,13,14,55,56]
**TIGIT**	+	+	+	-	[12,26,28]
**Nectin-1** (CD111, HVEC, PRR1, PVRL1, HIgR)	-	ND	-	ND	[36,39]
**Nectin-2** (CD112, HVEB, PRR2, PVRL2, PVRR2)	-	-	-	+	[36,40]
**Nectin-3** (CD113, PPR3, PRR3, PVRL3, PVRR3)	+	ND	+	+ *	[36,39]
**Nectin-4** (PVRL4, PRR4)	-	-	ND	-	[36,37,40]
**Necl-1** (CADM3, SYNCAM3, IGSF4B, TSLL1)	ND	ND	ND	ND	/
**Necl-2** (CADM1, SYNCAM1, IGSF4, IGSF4A, TSLC1, RA175)	+/-	-	ND	-	[52,53,54]
**Necl-3** (CADM2, SYNCAM2, IGSF4D)	ND	ND	ND	ND	/
**Necl-4** (CADM4, SYNCAM4, IGSF4C, TSLL2)	ND	ND	ND	ND	/
**Necl-5** (CD155, HVED, PVR, PVS, TAGE4)	+	+	ND	+	[40,41,44,57]

ND = not determined; + = present; - = absent; * = indirectly shown [36].

## 3. Effector Functions

To combat pathogens, immune effector functions are triggered upon Ag or target cell recognition, and need sufficient co-stimulation provided by the cellular environment. The balance between activating and inhibitory receptors controls immune homeostasis. However, dysregulation of these pathways contributes to chronic inflammation and autoimmune responses, leading to tissue destruction and, ultimately, immune senescence. In the following sections, we provide an overview on the immune-cell effector functions triggered by DNAM-1, CRTAM, Tactile and TIGIT and how this is altered in autoimmune or chronic inflammatory disorders (Figure 1). Herein, expression of these receptors can intrinsically affect effector functions, which are enhanced after ligand binding.

### 3.1. DNAM-1/CD226

DNAM-1 contributes to a series of innate and adaptive immune responses, including T- and NK-cell activation, proliferation, differentiation and effector functions. Herein, DNAM-1 is crucial in the formation of a stable IS of T lymphocytes and NK cells with ligand-positive APCs and target cells, respectively (Figure 1) [58,59]. Upon Ag or target cell recognition, DNAM-1 triggering elicits a co-stimulatory signal and promotes proliferation by ERK and Akt phosphorylation [8,14,59]. This is augmented when DNAM-1 co-localizes with LFA-1 and protein tyrosine kinase Fyn at the IS, thereby supporting granule polarization in NK cells [59,60,61]. DNAM-1 deficiency induces IS defects characterized by reduced LFA-1 recruitment and conjugate formation, and consequently impaired activation, proliferation and effector function [44,58,59].

In CD4^+^ T lymphocytes, DNAM-1 co-stimulation promotes pro-inflammatory Th cell differentiation, i.e., Th1 and Th17 cells, while suppressing Th2 cell and Treg differentiation by affecting transcription factor expression and cytokine production (Figure 1) [44,62,63,64]. Moreover, expression and stimulation of a DNAM-1 variant (i.e., a glycine-to-serine (G307S) SNP mutation associated with autoimmune disease susceptibility) in human CD4^+^ T lymphocyte increases their activation, proliferation and TNF-α/IFN-γ production, compared to WT DNAM-1 [64]. Accordingly, adoptive transfer of CD4^+^ T lymphocytes carrying this gain-of-function DNAM-1 mutation worsens EAE severity, which can be explained by increased pro-inflammatory T cells infiltrating the CNS [64]. Hence, DNAM-1-deficient T lymphocytes show reduced T-bet, RORγt, IFN-γ and IL-17 expression while GATA-3 or Foxp3 levels are upregulated, accompanied by increased IL-4, IL-13 or IL-10 production, leading to a Th2 or Treg phenotype in vitro [44,62,63]. Based on the effect on T helper cell balance, DNAM-1 deficiency ameliorates neuroinflammation in EAE, which is mainly attributable to increased Treg levels with upregulated suppressive capacity and decreased Th17-cell frequencies [63,65].

In NK cells, CD8^+^ and CD4^+^ CTLs (defined as CD4^+^CD28^null^ T cells), DNAM-1 expression is correlated to their cytotoxic capacity (Figure 1). Necl-5 expression on target cells is required for eliciting cytotoxicity in NK cells, indicating a targeted killing function [8,19,59]. Similarly, DNAM-1^+^ CD8^+^ T lymphocytes show high IFN-γ and granzyme B production which supports their killing function towards Necl-5^+^ endothelium in the context of SSc (Table 2) [17]. In addition, recruitment of DNAM-1 and LFA-1 at the IS features mature NK cells with high cytolytic potential [61,66]. Hence, DNAM-1- NK cells show poor LFA-1 expression, limited killing function and low IFN-γ production [66]. The latter is the case in MS patients, who show reduced DNAM-1 expression on NK cells and impaired Ag-induced Necl-5 upregulation on CD4^+^ T lymphocytes. Due to a reduced DNAM-1–Necl-5 interaction, NK cells cannot exert their effector function towards autoreactive T lymphocytes which results in defective immune regulation (Table 2) [19]. In EAE, NK cell-specific upregulation of DNAM-1 improves their immunomodulatory function by repressing Ag presentation of Necl-5^+^ DCs and enhances their cytotoxic function towards autoreactive CD4^+^ T lymphocytes [67]. In RA, DNAM-1^+^ NK cells are abundantly present in the peripheral blood and the synovium, where they attack Necl-5^+^ ICAM-1^+^ synovial fibroblasts (Table 2) [68]. In addition, the majority of CD4^+^ CTLs in the blood of RA patients show DNAM-1 expression, co-expressed with the activating receptor 2B4. Here, triggering of both DNAM-1 and 2B4 using agonistic antibodies elicits increased CD4^+^ CTL degranulation and IFN-γ secretion (Table 2) [69]. Although DNAM-1 variants are described as susceptibility genes for developing RA [22,24], DNAM-1 neutralizing antibody treatment or DNAM-1 deficiency in the collagen-induced arthritis (CIA) mouse model of RA did not affect disease development nor severity [70].

**Table 2 biology-12-00452-t002:** Overview of dysregulated and dysfunctional DNAM-1, CRTAM, Tactile and TIGIT in autoimmunity and chronic inflammation. Dysregulated expression profiles are based on human data. CRTAM, class-I-restricted T-cell-associated molecule; CTL, cytotoxic T lymphocyte; DNAM-1, DNAX accessory molecule 1; HIV, human immunodeficiency virus; Necl-2, Nectin-like protein 2; NK, natural killer; Tactile, T-cell-activated increased late expression; TIGIT, T-cell immunoreceptor with Ig and ITIM domains; Treg, T-regulatory cell.

Disease	Dysregulation	Dysfunction	Ref.
Multiple sclerosis	DNAM-1 ↘ on NK cells	Defective regulation of autoreactive T cells	[19,67]
	DNAM-1 ↗ on Tregs	Inducing a pro-inflammatory phenotype and reduced suppressive capacity	[20,44,62,63,64]
	TIGIT ↘ on B cells	Defective regulation of Th cell balance	[28,71]
Systemic sclerosis	DNAM-1 ↗ on CD8^+^ CTLs	Increased cytokine production and cytotoxicity	[17]
	DNAM-1 ↘ on NK cells	Defective regulation of autoreactive T cells	[21]
Rheumatoid arthritis	DNAM-1 ↗ on NK cells	Increased cytotoxicity towards synovial fibroblasts	[68]
	DNAM-1 ↗ on CD4^+^ CTLs	Increased cytokine production and cytotoxicity	[69]
	CRTAM ↗ in synovium	Hub gene for diagnosis and therapy	[72]
Type 1 diabetes	CRTAM ≈ on NK T cells and CD8^+^ CTLs	Targeted cytotoxicity of CD8^+^ CTLs towards Necl-2^+^ pancreatic islet cells	[32,73,74,75]
Atopic dermatitis	TIGIT ↗/↘ on CD4^+^ T cells	Retrains inflammation by inhibiting cell proliferation; negatively correlated with disease severity	[27]
Psoriasis	TIGIT ↘ on CD4^+^ T cells	Reduced regulation of cell proliferation and cytokine production; negatively correlated with disease severity	[29]
Ankylosing spondylitis	Tactile ↘ on CD4^+^ T cells	Inducing a pro-inflammatory phenotype and cytokine production	[18]
Viral infections	CRTAM ↗ on CD8^+^ CTLs	Increased cytotoxicity towards infected cells	[53,73,75]
	Tactile ↘ on CD8^+^ CTLs	Immunosenescent phenotype in persistent HIV infection	[76,77]

↘ = downregulated, ↗ = upregulated, ≈ not affected compared to healthy controls.

### 3.2. CRTAM/CD355

CRTAM regulates T-cell development and maturation, T-cell polarity, differentiation, cytotoxicity and cytokine production. Before entering the peripheral circulation, thymocytes mature from CD4^−^CD8^−^ to CD4^+^CD8^+^ thymocytes and eventually single positive CD4^+^ or CD8^+^ T lymphocytes. In mice, Necl-2 is expressed by thymocytes and CRTAM is constitutively expressed by CD4^-^CD8^-^ and single positive CD8^+^ thymocytes, which diminishes with age, indicating CRTAM expression is required during early thymocyte development. Consequently, CRTAM–Necl-2 interaction inhibition drastically impairs fetal thymus growth [54,78]. In the periphery, activated Ag-specific T lymphocytes mature into effector cells after APC stimulation in secondary lymphoid organs. Necl-2 is highly expressed by DCs and accordingly induces the strongest CRTAM upregulation compared to other APCs [32,34,54,75,79]. Moreover, CRTAM functions in CD8^+^ T-lymphocyte retention in the lymph nodes by supporting the IS with DCs. Therefore, the CRTAM–Necl-2 interaction is crucial for the accumulation of mature cytotoxic lymphocytes [34,75].

CRTAM gene expression is transcriptionally regulated, depending on the cellular activation state. While its expression is repressed by ZEB1 in resting and activated T lymphocytes, CRTAM is positively regulated by the AP-1 transcription factor in activated cells [80,81]. In turn, AP-1 regulates the expression of cytotoxicity-related genes, e.g., perforin and granzyme B [81]. Upon activation, CRTAM is upregulated and promotes cytotoxicity and IFN-γ production by NK cells and CD8^+^ T lymphocytes, preferably lysing Necl-2^+^ expressing cells, suggesting a targeted effector function (Figure 1) [31,32,53,82]. Although CRTAM triggering enhances cytotoxicity, CRTAM deficiency does not impair the effector function of NK cells and CD8^+^ CTLs, indicating a redundant role [31,75,83]. In this context, the CRTAM–Necl-2 interaction mainly supports IS formation with target cells which facilitates engagement of additional activating receptors [31].

In contrast to NK cells and CD8^+^ T lymphocytes, CRTAM expression on CD4^+^ T lymphocytes skews them into the cytotoxic T-lymphocyte lineage and is indispensable for their effector function (Figure 1) [34]. Upon T-cell activation, CRTAM upregulates IFN-γ, IL-17 and IL-22 production through Scrib signaling, and induces expression of eomesodermin, perforin and granzyme B [11,34]. The effect on cytokine production is mediated intrinsically by CRTAM, but is augmented by Necl-2 binding [11,34]. While Takeuchi et al. describe a Th1 phenotype for CD4^+^CRTAM^+^ T cells, because of spontaneous differentiation to IFN-γ producing cells, CRTAM expression is required for IL-17 production and Th17 response during intestinal parasitic or bacterial infections, where Necl-2 is expressed by gut-resident DCs [34,84,85,86]. After target cell triggering, CD4^+^CRTAM^+^ T lymphocytes elicit a cytotoxic response comparable to CD8^+^ CTLs, while stimulation with recombinant Necl-2 enhances their cytokine production [11,34]. When intracellular CRTAM signaling is abrogated in vitro or in vivo, CD4^+^ T lymphocytes do not acquire effector functions, indicating its crucial role in this subset [34]. The majority of CD4^+^CRTAM^+^ T lymphocytes exhibit an EM phenotype. Accordingly, CRTAM transgenic mice show increased EM T cells, associated with elevated effector cytokines levels, while Necl-2-deficient mice show a 10-fold increase in naive T lymphocytes, suggesting that the CRTAM–Necl-2 interaction is involved in EM differentiation [11,34,54]. This is partly mediated by the intracellular interaction of CRTAM and Scrib which polarizes CRTAM and CD3 asymmetrically to CD44 after activation [11].

These CRTAM-induced effector functions are essential in restraining microbial infections and contribute to (auto-)inflammatory responses (Table 2) [11,34,75]. Herein, human viral antigen-responsive CD8^+^ T cells (i.e., influenza, CMV) selectively upregulate CRTAM as distinct marker, compared to control-stimulated CD8^+^ T cells [73]. Interestingly, human T-cell lymphotropic virus-1 (HTLV-1)-infected CD4^+^ T lymphocytes upregulate Necl-2, which marks them for CD8^+^ CTL-mediated lysis and promotes elimination of infected cells [53]. In vivo, CRTAM-deficient mice show reduced protective immunity against influenza, due to a reduced migration of influenza-specific CD8^+^ CTLs and diminished cytotoxic function of CD4^+^ T cells [34,75]. In addition, CRTAM^+^CD4^+^ T cells contribute to the induction of T-cell-mediated intestinal colitis in mice, since adoptive transfer of CRTAM-deficient T cells failed to induce gut inflammation [34]. Moreover, CRTAM-deficient mice show an impaired induction of CD8^+^ T-cell-mediated autoimmune diabetes [75]. In pancreatic tissue of T1D patients, Necl-2 is highly expressed by pancreatic islet cells and myeloid cells in the proximity of CD8^+^ T lymphocytes, which suggests their susceptibility for CRTAM-mediated cytotoxicity and promotes T-cell–APC interactions, respectively [74]. However, human T1D autoantigen-responsive CD8^+^ T cells (i.e., responsive to islet-specific glucose-6 phosphatase catalytic subunit-related protein) do not show a distinct CRTAM expression profile compared to control-stimulated cells, and no differences on NK T cells were detected in T1D patients, compared to healthy subjects [32,73]. While CRTAM expression seems unaffected in human T1D, it is identified as hub gene for diagnosis and therapy of RA patients since its expression is significantly upregulated in the synovial tissue of RA patients and rodents, although this is not cell-specific [72]. For the role of the CRTAM–Necl-2 interaction in tumor immune escape mechanisms, see [87,88,89].

### 3.3. Tactile/CD96

The role of Tactile in regulating lymphocyte effector functions is rather controversial since it depends on the stimulus and its murine or human origin. Tactile displays a cytoplasmic ITIM motif, giving it the ability to transduce inhibitory signals [10]. While Tactile promotes CD8^+^ T-lymphocyte activation and proliferation, cytokine production and cytotoxicity in mice, it limits these functions in human cells [14,76]. Murine Tactile acts as a co-stimulatory signal in CD4^+^ and CD8^+^ T lymphocytes by inducing activation and proliferation through MEK–ERK signaling, comparable to CD28 (Figure 1) [14]. This co-stimulation enhances the CD8^+^ T cell’s cytotoxic response towards target cells, preferentially lysing Nectin-1^+^ and Necl-5^+^ cells [14]. Hence, Tactile blockade on CD8^+^ T lymphocytes impairs Ag-induced activation and IFN-γ production and migration towards inflammatory tissue in vivo, after ovalbumin triggering [14]. Likewise, Tactile blockade or deficiency in a mouse model of psoriasis results in milder dermatitis symptoms, which is attributable to a reduced accumulation, activation and IL-17 production of dermal γδ T cells. Herein, Tactile also functions as a skin-homing marker since Necl-5 is highly upregulated on inflamed skin tissue [90]. In contrast, Tactile limits the antitumor response of CD8^+^ T lymphocytes in the context of cancer, by negatively affecting its effector functions in vivo [91]. Moreover, the inflammatory potential of murine Th9 cells is negatively correlated to Tactile expression since high IL-9 producers are associated with low Tactile expression and vice versa. Consequently, adoptive transfer of Tactile^Low^ Th9 cells into immune-compromised mice induces severe intestinal and skin inflammation, while Tactile^High^ Th9 cells do not cause pathology [92].

In humans, activated Tactile^+^ CD8^+^ T cells lack perforin expression, while Tactile- CD8^+^ T lymphocytes are highly cytotoxic, characterized by perforin and IFN-γ secretion [76]. Similarly, Tactile downregulation in human CD4^+^ T cells is associated with a pro-inflammatory phenotype while its overexpression inhibits an inflammatory response by affecting cytokine production of IL-17, TNF-α, IL-23, IL-6, IFN-γ, and ERK signaling [18]. Accordingly, Tactile expression is reduced on CD4^+^ T cells of patients with ankylosing spondylitis, a chronic inflammatory type of arthritis (Table 2) [18]. Hence, the activation state of T lymphocytes is negatively correlated to Tactile expression, indicating an immunosuppressive function which is inconsistent with mouse Tactile (Figure 1). However, antibody-mediated inhibition of Tactile on human CD8^+^ T lymphocytes neither increases nor decreases IFN-γ production and cytotoxic capacity, whereas blocking the inhibitory receptor TIGIT increases effector functions, suggesting that Tactile is not a true inhibitory receptor [93]. In addition, persistent HIV infection induces an IL-32-mediated Tactile downregulation on CD8^+^ T cells which triggers an immunosenescent phenotype, characterized by reduced expression of co-stimulatory molecules and lower proliferation, leading to a suboptimal control of viral antigens (Table 2) [76,77]. Finally, in Crohn’s disease, genetic polymorphisms of Tactile have been associated with treatment unresponsiveness to TNF-α inhibitors, i.e., adalimumab [94]. Interestingly, healthy control-derived CD4^+^ T cells subjected to TCR-stimulation and treated with adalimumab will upregulate Tactile expression [95]. Since TNF-α inhibitors are widely used as first-line therapy in autoimmune and inflammatory diseases, this genetic susceptibility for treatment failure might be translatable to other disorders and can contribute to patient-specific predictions of treatment responsiveness [94,95].

In NK cells, mouse Tactile limits its effector functions, while human Tactile promotes adhesion and cytotoxicity [39,55,96]. Tactile-deficient mice exhibit a detrimental hyperinflammatory response against LPS stimulation, associated with a boost in IFN-γ-producing NK cells, while being more resistant to tumor development and metastasis [96]. In human NK cells, Tactile binding to its ligands on target cells promotes NK cell adhesion and a targeted cytotoxic response [39,55]. Moreover, Tactile does not halt the cytotoxic response when co-engaged with NK-cell-activating receptors, indicating that it does not transduce inhibitory signals despite the cytoplasmic ITIM motif [55]. However, the killing capacity of Tactile-triggered NK cells by Necl-5^+^ target cells is less effective than DNAM-1 or other NK-cell-activating receptor stimulation, suggesting that Tactile-mediated cytotoxicity requires additional stimuli [55]. Even though Tactile promotes targeted cytotoxicity in human NK cells, it functions presumably in mediating a mature IS between effector and target cell, i.e., through Tactile–Nectin-1 and –Necl-5 interactions, which promotes engagement of additional activating signals (Figure 1) [39,55]. Interestingly, patients with chronic obstructive pulmonary disease (COPD) show increased frequencies of circulating Tactile^+^ NK cells after an acute exacerbation, although the overall amount of NK cells is reduced in patients [97]. If Tactile ligands Nectin-1 and/or Necl-5 are highly expressed in lungs, this suggests that Tactile expression may function as a biomarker for COPD progression.

### 3.4. TIGIT

TIGIT carries ITT-like and ITIM motifs on its cytoplasmic domain by which it suppresses T-cell activation, proliferation, and differentiation and exerts regulatory effector functions through a shift in the balance of cytokines [9,29,44]. Additionally, it is characterized as an immunological check-point inhibitor and marker for T-cell exhaustion [98]. TIGIT can act indirectly as well as directly on the T-cell cycle. TIGIT interacts with its predominant ligand Necl-5 on the surface of mature DCs in order to increase their secretion of IL-10 and reduce the secretion of IL-12, thereby inducing a immunoregulatory DC phenotype and indirectly inhibiting T-cell activation [26]. Herein, the MAPK–ERK pathway, that has a critical role in IL-10 modulation, is induced after Necl-5 phosphorylation following TIGIT ligation [26]. This effect can be reversed by IL-10 neutralization and ERK inhibition, demonstrating the presence of an IL-10-driven feedback loop [26]. On the other hand, TIGIT expression on T lymphocytes acts directly and independent of APCs by affecting CD4^+^ T-cell proliferation and Th1 and Th17 responses via IL-10 secretion and the suppression of TCR- and CD28-driven signaling pathways (Figure 1) [26,29,44,99,100]. This is accompanied by a suppression of transcription factors T-bet, RORc, GATA3 and IRF4, associated with the regulation of Th1, Th17, Th2 and Th9 [44]. Accordingly, CD4^+^ T-cell proliferation and the levels of IFN-γ and IL-17 along with T-bet are enhanced, while IL-10 levels are downregulated following a TIGIT knockdown [29,44]. Herein, the increased production of IFN-γ can be reversed by the blockage of DNAM-1 signaling, as TIGIT shares the DNAM-1 ligands Necl-5 and Nectin-2, forming a TIGIT/DNAM-1 axis [44,101]. Furthermore, DNAM-1 deficiency enhances TIGIT signaling which preserves Foxp3 expression and Treg function under inflammatory conditions, in a mouse model of acute graft-versus-host disease [101]. Hence, TIGIT carries out immunosuppressive effects by competing with DNAM-1. Consequently, TIGIT-deficient mice are more prone to EAE due to a hyperproliferative T-cell response, while TIGIT overexpression on CD4^+^ T cells in a mouse model of RA alleviates disease severity, supporting the inhibitory function of TIGIT and its role in the regulation of autoimmune responses through the TIGIT/DNAM-1 axis [29,65,99,100].

NK and CD8^+^ T-cell cytotoxicity is downregulated by TIGIT–ligand binding on target cells (Figure 1) [9,102]. In contrast to DNAM-1, TIGIT directly inhibits cytotoxicity in human and murine NK cells since TIGIT–Necl-5 interaction is dominant over DNAM-1–Necl-5 binding [103]. Herein, TIGIT triggering by Necl-5^+^ target cells results in the disruption of granule polarization through the recruitment of SHIP1 by its ITT-like motif [3,9,103]. Similarly, SHIP1 recruitment to TIGIT on CD8^+^ CTLs results in reduced secretion of granzyme B, TNF-α and IFN-γ due to inhibition of ERK, MAPK and transcription factor NF-κB [104]. Blockage of the TIGIT–Necl-5 interaction restores the impaired NK cell and CD8^+^ CTL effector function. This mechanism is often used by tumors, which upregulate Necl-5 and Nectin-2, to evade a cytotoxic response [104,105]. TIGIT can allegedly also bind to Nectin-3, however, there are currently no clear effector functions described for this interaction [103]. Interestingly, since epithelial cells express Necl-5, Nectin-2 and Nectin-3, it is believed that this generates additional protection against self-destruction by NK cells [103]. Lastly, elevated TIGIT expression can be found on exhausted CD8^+^ T cells [98]. Herein, TIGIT is seen as a critical regulator of cell exhaustion, associated with poor clinical outcome of cancer and chronic viral infections [98,106,107]. Controversially, exhaustion of CD8^+^ T cells has been shown to predict a more positive outcome for autoimmunity [106,108].

Recently, TIGIT expression was also identified on human memory B cells [28,71]. In healthy control B cells, IL-4 is able to downregulate TIGIT expression after B-cell activation by affecting TCF4 transcription factor expression [71]. TIGIT^+^ B cells can directly suppress T-cell responses in vitro by repressing IFN-γ and IL-17 production. In addition, they affect Necl-5^+^ DC maturation, which indirectly results in a reduced Th-cell response and promotes Treg-mediated immune regulation through IL-10 [28]. In MS patients, TIGIT expression is impaired on memory B cells, indicating reduced immune regulation. In this context, the impaired TIGIT expression seems to drive Th17-like follicular Th-cell expansion, pointing towards a dysregulated loop resulting in continued activation of the immune system (Table 2) [71]. However, the role of TIGIT in B cell function needs further investigation.

## 4. Conclusions

In this review, we described how Nectin-family ligands modulate immune effector functions in health, chronic inflammation and autoimmunity. Herein, NK cells and T lymphocytes are the main cells expressing DNAM-1, CRTAM, Tactile, TIGIT and Nectin-3 in the innate and adaptive immune systems. However, the expression patterns of other Nectins and Necls are rather incomplete. DNAM-1- and CRTAM-triggering will promote a pro-inflammatory immune response since they function as co-activating receptors that boost Th1- and Th17-cell cytokine production, while they enhance targeted cytotoxicity towards Necl-5^+^ and Necl-2^+^ cells by NK cells and CD8^+^ and CD4^+^ CTLs. In addition, DNAM-1 and CRTAM interaction with their ligands strengthens the IS with APCs and target cells, e.g., Necl-5 is highly expressed by DCs and activated CD4^+^ T cells. In contrast, TIGIT and Tactile act as co-inhibitory receptors in human cells. TIGIT triggering induces an immunoregulatory phenotype in DCs, T cells, B cells and NK by boosting IL-10 secretion. Interestingly, TIGIT and DNAM-1 share the same ligands, Necl-5 and Nectin-2, which results in competing cellular responses, indicating that a balance between these receptors is crucial for immune homeostasis. On the other hand, Tactile–ligand interaction supports cell adhesion at the IS and regulates immune responses by inhibiting CD8^+^ CTL cytotoxicity and Th1- and Th17-cell differentiation. However, Tactile has opposite functions in mouse and human cells, which challenges translational research.

Finally, Nectin-family ligands are differentially expressed in NK cells, B cells and CD4^+^ and CD8^+^ T lymphocytes during chronic inflammation and autoimmunity. Although altered expression profiles of DNAM-1, CRTAM, Tactile and TIGIT are reported in peripheral blood samples of patients diagnosed with MS, RA, SSc, psoriasis, etc., a correlation with disease severity and progression in time is often lacking. In addition, modulating the DNAM-1/TIGIT axis is highly investigated as a therapeutic target in oncoimmunity, and can also be of interest to target autoimmunity to boost immunoregulation and reduce target-cell killing [104,105]. Furthermore, genetic variants of DNAM-1 [22], CRTAM [72] and Tactile [94] are identified as being related to autoimmune susceptibility, the hub gene for diagnosis and unresponsiveness to immunosuppressive therapies, which underscores the importance of further genome-wide association studies in this field. In conclusion, Nectin-family ligands create an interesting scientific outlook to understand dysfunctional immune responses in chronic inflammatory disorders and autoimmune diseases.

## Figures and Tables

**Figure 1 biology-12-00452-f001:**
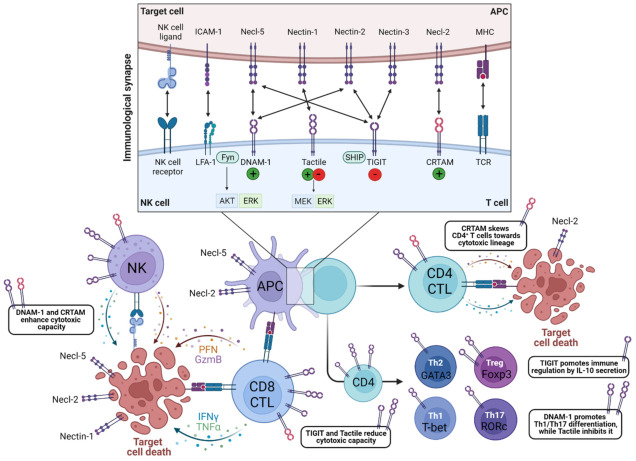
Overview of DNAM-1-, CRTAM-, Tactile- and TIGIT-triggered immune effector functions. The immunological synapse (IS) is formed between a T lymphocyte and APC or between cytotoxic cells and their target cell, as a result of TCR/MHC interaction or NK cell receptor binding, enabling long-lasting cell contact. DNAM-1 co-localizes with LFA-1 and protein kinase Fyn at the IS, which is crucial in the formation of a stable IS and supports granule polarization in cytotoxic cells. Besides enhancing a targeted cytotoxic response towards ligand-expressing target cells, DNAM-1 contributes to the activation, proliferation and differentiation of Th1 and Th17 cells. CRTAM is transiently expressed by activated T cells and NK cells and regulates pro-inflammatory cytokine production and cytotoxicity. CRTAM expression skews CD4^+^ T cells towards a cytotoxic lineage. CRTAM–Necl-2 interaction supports IS formation and a targeted cytotoxic response. On the contrary, Tactile expression reduces the cytotoxic function of CD8^+^ CTLs, suppresses a pro-inflammatory phenotype of CD4^+^ T cells and supports NK cell adhesion at the IS. TIGIT carries out immunosuppressive effects by competing with DNAM-1 for its ligands, leading to suppression of proliferation, differentiation, cytotoxicity by increasing the production of IL-10. APC, antigen presenting cell; CRTAM, class-I-restricted T-cell-associated molecule; CTL, cytotoxic T lymphocyte; DNAM-1, DNAX accessory molecule 1; GzmB, granzyme B; ICAM-1, intercellular adhesion molecule 1; IFNγ, interferon gamma; IL, interleukin; LFA-1, Lymphocyte function-associated antigen 1; MHC, major histocompatibility complex; NK, natural killer cell; Necl, Nectin-like molecule; PFN, perforin; Tactile, T-cell-activated increased late expression; TCR, T-cell receptor; Th, T helper cell; TIGIT, T-cell immunoreceptor with Ig and ITIM domains; TNFα, tumor necrosis factor alpha. Created with Biorender.com (accessed in January 2023).

## Data Availability

Not applicable.

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
