# Peer review of "Nectin Family Ligands Trigger Immune Effector Functions in Health and Autoimmunity"

_biology, 2023, doi:10.3390/biology12030452_

Round 1

Reviewer 1 Report

In this review article Doryssa Hermans et.al tried to cover an important interaction between the nectin family and its receptor present in different immune cell types and their role in disease and autoimmunity. The authors need to put more information as there are some important pieces of information missing.

Major Suggestions: 

  1.  Line (17-20)The first sentence of the abstract is too long and difficult to read, authors might consider rephrasing it. You would not like readers to drop their enthusiasm in the first line of your abstract. I will try to make it simple and attractive.
  2. This statement is highly redundant and appears 3 times (lines 17-20, lines 44-48, and lines 66-67) in just 2 pages. Authors can consider reframing it.
  3. Table 1 needs revision. I would suggest including macrophage cells and looking out for other synonyms of the genes to fill out the table accurately. Macrophages also present antigens in autoimmune conditions such as Type-1 diabetes. (Ref: Evidence of islet CADM1-mediated immune cell interactions during human type 1 diabetes). CADM1 is a synonym of Necl-2. The table has most of the things undetermined, if authors focus on other synonyms names of these genes and consider immune cells like macrophages etc. would help look more informative.
  4. Necl-2 also known as CADM1 is not addressed. I will like authors to introduce and use other popular synonymous names. There are papers for CADM1 and CRTAM interaction these all need to be cited.
  5.  TIGIT is an immunological checkpoint inhibitor and marker for T-cell exhaustion. While introducing TIGIT the authors should have these pieces of information in the text as it's a review article. TIGIT's role in T-cell exhaustion is an important aspect of autoimmunity. This information is missing to make it more interesting for the audience.
  6. While referring to studies information remains vague. The review is about ligand and receptor interaction. Thus while referring to studies in many places it will be nice to describe where the ligands are expressed. Further, these ligands are also expressed in non-immune cells also, and how these modulate the effector function of immune cells also should be discussed in the review.
  7. Further, after modifying based on synonymous names, authors would also consider modifying their figure 1.
  8. The supplementary information attached is sequencing data from other projects and needs to be removed, not relevant to this study. If authors did not upload these files, I would say it would be a software glitch. 

Author Response

Reviewer 1:

In this review article Doryssa Hermans et.al tried to cover an important interaction between the nectin family and its receptor present in different immune cell types and their role in disease and autoimmunity. The authors need to put more information as there are some important pieces of information missing.

We thank the reviewer for critically assessing our manuscript. The comments of this reviewer are addressed below in a point by point reply.

Major Suggestions:

  1. Line (17-20) The first sentence of the abstract is too long and difficult to read, authors might consider rephrasing it. You would not like readers to drop their enthusiasm in the first line of your abstract. I will try to make it simple and attractive.

We agree with the reviewer that this sentence it too long due to the complete names of the discussed proteins (DNAM-1, CRTAM, Tactile, TIGIT). Therefore, we shortened and split the sentence (page 1; lines 16-21). However, we cannot remove the complete terms and abbreviations.

  1. This statement is highly redundant and appears 3 times (lines 17-20, lines 44-48, and lines 66-67) in just 2 pages. Authors can consider reframing it.

As requested by the reviewer, we rephrased the sentence from comment 1 (page 1; lines 16-21). We agree with the reviewer that lines 44-48 and lines 66-67 accidently repeat themselves. Therefore, we reframed one of these sentences (page 2; lines 71-73).

  1. Table 1 needs revision. I would suggest including macrophage cells and looking out for other synonyms of the genes to fill out the table accurately. Macrophages also present antigens in autoimmune conditions such as Type-1 diabetes. (Ref: Evidence of islet CADM1-mediated immune cell interactions during human type 1 diabetes). CADM1 is a synonym of Necl-2. The table has most of the things undetermined, if authors focus on other synonyms names of these genes and consider immune cells like macrophages etc. would help look more informative.

As correctly stated by the reviewer, different synonyms of Nectins and Necls are used in literature. To make this clear for the reader, we added all the synonyms used for the human proteins in table 1 (page 4; lines 126-134). In addition, we clarified why different nomenclature is used for Nectins and Necls in the introduction (page 2; lines 53-55). We did include all synonyms in our search while generating table 1. However, table 1 summarizes research on (healthy) human immune cells (based on peripheral blood mononuclear cells (PBMCs) or NK-92 and Jurkat T cell lines) and on protein expression levels which is why some proteins are not reported yet in literature (= undetermined). In addition, we discussed controversial literature on Necl-2 expression in the immune system and included this in table 1 (page 3; lines 117-124). As suggested by the reviewer, we broadened our search to macrophages, leading to minor changes of the table content. By including the ‘undetermined’ expression profiles, this review also underscores the current research gaps.

  1. Necl-2 also known as CADM1 is not addressed. I will like authors to introduce and use other popular synonymous names. There are papers for CADM1 and CRTAM interaction these all need to be cited.

As mentioned in our response to comment 3, we included all the synonyms used for Nectins/Necls in table 1 (page 4). The CRTAM-Necl-2/CADM1 interaction is discussed in the section of CRTAM effector functions (pages 6-7). We did include all Necl-2 synonyms in our search. However, after revising the current literature, we added additional references that fit the scope of our review (Kim HR. et al., J Exp Med., 2011 & Manivannan K. et al., PLoS Pathog., 2016), focussing on immune cell interaction in chronic inflammation and autoimmunity (page 7; lines 242-245); excluding oncoimmunity. As suggested by the reviewer in comment 3, we also included Sona C. et al., JCI Insight, 2022 in the discussion of the CRTAM-Necl-2 interaction (page 7; lines 251-254). Considering literature on this interaction in the field of cancer immunology, we referred the reader to other papers and reviews (page 7; lines 260-261).

  1. TIGIT is an immunological checkpoint inhibitor and marker for T-cell exhaustion. While introducing TIGIT the authors should have these pieces of information in the text as it's a review article. TIGIT's role in T-cell exhaustion is an important aspect of autoimmunity. This information is missing to make it more interesting for the audience.

As well-noticed by the reviewer, TIGIT is also described as immunological checkpoint inhibitor and marker for T-cell exhaustion. We now included this point in the discussion of TIGIT (page 8, lines 332-333; page 9, lines 371-374). Since this TIGIT function is mainly applicable to oncoimmunity and viral infections, we referred the reader to the according literature.

  1. While referring to studies information remains vague. The review is about ligand and receptor interaction. Thus while referring to studies in many places it will be nice to describe where the ligands are expressed. Further, these ligands are also expressed in non-immune cells also, and how these modulate the effector function of immune cells also should be discussed in the review.

We agree with the reviewer that the effector functions triggered by DNAM-1, CRTAM, Tactile and TIGIT are complex. These receptors can act directly on immune cell functions, independent of ligand binding (intrinsic pathways), while ligand binding often augments the functional outcome. We clarified this statement (page 5; lines 143-144) and included more details on in vitro studies, where often target cells are used in which ligand expression was induced by transfection (page 5, line 172; page 6, lines 189, 198, 221-230; page 8, lines 294-295, 317; page 9, line 361). Considering in vivo studies, ligand-expressing cells are mentioned when it is known from the according studies.

  1. Further, after modifying based on synonymous names, authors would also consider modifying their figure 1.

As mentioned in our response to comment 3, we included all the synonyms used for Nectins/Necls in table 1 (page 4). Since we included all synonyms in our literature search of the submitted manuscript, the content of the review does not need to be modified. We decided to not include all the Nectin/Necl synonyms in figure 1, as this would complicate the overview.

  1. The supplementary information attached is sequencing data from other projects and needs to be removed, not relevant to this study. If authors did not upload these files, I would say it would be a software glitch.

We did not attach sequencing data with our submission. This would, indeed, be a software glitch.

Reviewer 2 Report

The authors have successfully explained how the Nectin family ligands modulate immune effector functions in health, chronic inflammation and autoimmunity. I would like the authors to add one figure to describe how the deregulation of different Nectin ligands lead to different autoimmune diseases through a figure. I find the paper well written and good enough to explain the essential role of Nectin family.

Author Response

The authors have successfully explained how the Nectin family ligands modulate immune effector functions in health, chronic inflammation and autoimmunity. I would like the authors to add one figure to describe how the deregulation of different Nectin ligands lead to different autoimmune diseases through a figure. I find the paper well written and good enough to explain the essential role of Nectin family.

We would like to thank Reviewer 2 for their acknowledgment of our review’s value. As suggested by the reviewer, we added an overview on the dysfunction/dysregulation of DNAM-1, CRTAM, Tactile and TIGIT in autoimmune disorder and chronic inflammation, focussing human data. However, we decided to summarize this in a table, instead of a figure, to add the correct references (Table 2; page 10; lines 388-395).

Round 2

Reviewer 1 Report

I will like to thank the authors for incorporating my suggestions. The article will be good read for people interested in interaction between the nectin family and its receptor present in different immune cell.